# Participation and adherence to mammography screening in the Capital Region of Denmark: The importance of age over time

**Lindsay Pett**[ID]◎*, **Becky Hollenberg**[ID]◎, **Jessica Mahoney**◎, **Jake Paz**◎, **Nathan Siu**◎, **Amanda Sun**◎, **Rachel Zhang**◎, **My von Euler-Chelpin**◎

Department of Public Health, University of Copenhagen, Copenhagen, Denmark

◎ These authors contributed equally to this work.
* Lindsay.pett@richmond.edu

**Data Availability Statement:** The data underlying this article cannot be shared for the privacy of the individuals who participated in the mammography screening programs. The data will be shared on

## Abstract

Mammography screening's effectiveness depends on high participation levels. Understanding adherence patterns over time is important for more accurately predicting future effectiveness. This study analyzed longitudinal adherence to the biennial invitations in the Capital Region of Denmark from 2008–2017. We analyzed participation rates for five-year age groups along with their percent changes in each invitation round using linear regressions. Participation in the mammography screening program increased from 73.1% to 83.1% from 2008–2017. The participation rate among all age groups increased from the first to the fifth round, with the oldest age group having the largest increase (average percent change = 3.66; p-value = 0.03).

## Introduction

Breast cancer, worldwide, is the most common cancer in women and was responsible for more than 600,000 deaths in 2018 [1]. Additionally, breast cancer now causes the highest annual number of cancer deaths in women [1]. Studies have established that mammography screening is an effective tool for reducing breast cancer mortality [2–5]. However, its effectiveness depends upon high levels of participation in the screening program. The European Guidelines for Quality Assurance in Breast Cancer Screening and Diagnosis recommend that at least 70–75% of a population participate in regular mammography screening [6].

### Background

To achieve this goal of high participation, substantial research has been conducted to assess participation in Denmark and numerous other countries. There are personal, socio-economic, as well as cultural differences in adherence to organized mammography screening [2, 7–14]. Poorer physical and mental health, diminished primary care physician interaction levels, lower social support, lower educational attainment, and older age have been associated with lower

reasonable request to the corresponding author. The data are stored in the national bureau of statistics in Denmark, called Statistics Denmark (DST). The data is pseudo-identified, i.e. the Danish personal identification number is switched with another number. However, there is a key between the two which according to Danish law constitutes a risk for potential identification. The data can, with the relevant permissions, be accessed by contacting the Department of Public Health, University of Copenhagen, Denmark with reference to the research project "Benefits and harms of screening for breast cancer with mammography" and through that department potentially gain access to the data at DST. Department of Public Health Faculty of Health and Medical Sciences University of Copenhagen Phone: +45 35 32 76 23 Department contact: George Napolitano, gena@sund.ku.dk https://publichealth.ku.dk/.

**Funding:** The authors received no specific funding for this work.

**Competing interests:** The authors have declared that no competing interests exist.

levels of adherence [2, 7, 8, 10–14]. Several of these possible determinants are also independently related to age. In an earlier study from Denmark, the participation rate was significantly lower in the age group 65–69 than in the age group 50–54 (p<0.0001) [15]. Interventions targeting these groups may promote equal participation in future breast cancer screening programs. Although research has identified many factors that affect participation rates, many of these studies look only at singular screenings or screenings over a few years [7–14]. Relatively few studies have examined longitudinal adherence by age group across multiple invitation rounds in a national screening program. One longitudinal study from Lithuania examines screening participation rates from 2006–2014, noting that participation increased more than three-fold over this time period [16]. Another such study from Spain found that adherence rates decreased from 2011 to 2017 [17].

We wanted to assess the possible change in participation and adherence by age group over time. To this end, we used data from the Capital Region Mammography Screening Register to analyze time trends and longitudinal adherence to the biennial invitations in the capital region of Denmark from 2008–2017.

## Material and methods

Healthcare, and thereby also screening, is free of charge in Denmark. In the Capital Region, breast cancer screening is offered every second year to all women aged 50–69 years. The invitation is personal and women are invited with a fixed, changeable, appointment to visit one of the five mammography screening clinics in the region. Women who do not show up are sent a reminder. Participation, assessment of abnormal findings, and eventual treatment are all free of charge for the women. Women not wanting to participate can opt out of the invitation scheme. Each two-year invitation period is called an invitation round. Women diagnosed with cancer will be eligible for rescreening within the program after approximately 18 months.

For the present study, we retrieved data on all mammography screening examinations from 1 January 2008 to 31 December 2017 from the Capital Region Mammography Screening Register. Data was split by 5-year age groups and stratified by each invitation round. The study is entirely a register study, and no contact was made to the women included in the screening program. Consequently, no approval from the Ethical Committee or informed participant consent was needed according to Danish Law. The project was approved by The Faculty of Health and Medical Sciences under the General Data Protection Regulation, Regulation 2016/679, Ref. no.: 514-0238/18-3000. All data were pseudo-anonymized before use by the authors, and no images were accessed.

Data from this study are stored in Statistics Denmark, which only can be accessed given the relevant data permits in accordance with GDPR and Danish Law. The data is based on personal identification numbers and open access would seriously compromise the privacy of the women included in the study.

Participation was calculated as the number of women who participated in each invitation round divided by the number of women invited for each invitation round and stratified by 5-year age groups. The percent change in participation rates for the five participation rounds were calculated for each age group. The 95% confidence intervals for the participation rates were calculated using Vasser Stats's confidence interval of a proportion calculator [18]. Joinpoint Regression Program version 4.8.0.1 (2020) [19] was used to run linear regressions of the participation rates within each age group and analyze differences between the slopes. An alpha of p = 0.05 was used for all analyses.

## Results

Overall participation in the mammography screening program increased from 73.1% to 83.1% from the first invitation round to the final invitation round (Table 1). In invitation round one, 190,583 women were invited to participate in the mammography screening program. 73.1% of these women participated in screening [95% CI: 73.0, 73.4]. Among the age groups, participation rates ranged from 68.7% in the oldest age group [95% CI: 68.3, 69.2] to 75.2 among the 55–59 age group [95% CI: 74.9, 75.6]. In invitation round two, 185,966 women were invited to participate in the mammography screening program. 72.2% of these women participated in screening [95% CI: 72.1, 72.5]. Among the age groups, participation rates ranged from 71.1% in both the youngest and oldest age groups [Age 50–54 95% CI: 70.8, 71.6; Age 65–69 95% CI: 70.7, 71.5] to 74.3 among the 60–64 age group [95% CI: 74.0, 74.7]. The total percent change in participation rate from round one to round two was -1.23%, ranging from -3.86% in the 55–59 age group to 3.49% in the oldest age group. In invitation round three, 181,475 women were invited to participate in the mammography screening program. 79.9% of these women participated in screening [95% CI: 79.7, 80.0]. Among the age groups, participation rates ranged from 76.4% in the youngest age group [95% CI: 76.1, 76.8] to 82.2 among the 60–64 age group [95% CI: 81.9, 82.6]. The total percent change in participation rate from round two to round three was 10.7%, ranging from 7.5% in the youngest age group to 13.08% in the oldest age group. In invitation round four, 204,315 women were invited to participate in the mammography screening program. 77.7% of these women participated in screening [95% CI: 77.5, 77.9]. Among the age groups, participation rates ranged from 76.6% in the youngest age group [95% CI: 76.3, 77.0] to 79.3 among the 60–64 age group [95% CI: 78.9, 79.6]. The total percent change in participation rate from round three to round four was -2.75%, ranging from -3.95% in the 55–59 age group to 3.53% in the 60–64 age group. In invitation round five, 186,671 women were invited to participate in the mammography screening program. 83.1% of these women participated in screening [95% CI: 82.9, 83.2]. Among the age groups, participation rates ranged from 80.3% in the youngest age group [95% CI: 80.0, 80.6] to 84.7 among the 60–64 age group [95% CI: 84.3, 85.0]. The total percent change in participation rate from round two to round three was 6.95%, ranging from 4.83% in the youngest age group to 8.54% in the oldest age group.

Of the 91,988 women invited to all 5 rounds (i.e. women who were alive, living in the Capital region, accepting invitation and in the correct age group), 64.8% attended all five rounds. 32.8% of women attended some of the rounds, and 2.4% attended none of the rounds (Table 2).

The youngest age group had an average percent change (APC) of 1.87 with a p-value of 0.06, indicating that the APC was not significantly different from zero at the alpha = 0.05 level (Fig 1). The 55–59 age group had an APC of 2.40 with a p-value of 0.11, indicating that the APC was not significantly different from zero at the alpha = 0.05 level. The 60–64 age group had an APC of 2.58 with a p-value of 0.052, indicating that the APC was borderline significantly different from zero at the alpha = 0.05 level. The oldest age group was the only group that had a statistically significant average percent change in participation rate from the first invitation round to the final invitation round. The APC of 3.66 with a p-value of 0.03 indicated that the APC was significantly different from zero at the alpha = 0.05 level. The total regression had an APC of 2.55 with a p-value of 0.0496, indicating that the APC was significantly different from zero at the alpha = 0.05 level.

## Discussion

We found an increasing participation rate in all age groups over time. The most significant change occurred within the older age groups, who participated to a larger extent than they did

**Table 1. Mammography screening invitations and participation in the Capital Region of Denmark from 2008–2017 by age group.**

| | | Invitation Round 1 | | | Invitation Round 2 | | | | Invitation Round 3 | | | | Invitation Round 4 | | | | Invitation Round 5 | | |
| | | 2008–2009 | | | 2010–2011 | | | | 2012–2013 | | | | 2014–2015 | | | | 2016–2017 | | |
| | | Invited Population | Participated (% of invited) [95% CI] | | Invited Population | Participated (% of invited) [95% CI] | % Change | | Invited Population | Participated (% of invited) [95% CI] | % Change | | Invited Population | Participated (% of invited) [95% CI] | % Change | | Invited Population | Participated (% of invited) [95% CI] | % Change |
|---|---|---|---|---|---|---|---|---|---|---|---|---|---|---|---|---|---|---|---|
| AGE | 50–54 | 51833 | 38182 (73.7) [73.3, 74.0] | | 49571 | 35269 (71.1) [70.8, 71.6] | -3.53 | | 52634 | 40237 (76.4) [76.1, 76.8] | 7.45 | | 58656 | 44936 (76.6) [76.3, 77.0] | 0.26 | | 59012 | 47391 (80.3) [80.0, 80.6] | 4.83 |
| | 55–59 | 46649 | 35097 (75.2) [74.9, 75.6] | | 46156 | 33392 (72.3) [71.9, 72.8] | -3.86 | | 43774 | 35491 (81.1) [80.7, 81.4] | 12.17 | | 49007 | 38187 (77.9) [77.6, 78.3] | -3.95 | | 45582 | 38460 (84.4) [84.0, 84.7] | 8.34 |
| | 60–64 | 51045 | 37921 (74.3) [73.9, 74.7] | | 47447 | 35276 (74.3) [74.0, 74.7] | 0 | | 41200 | 33885 (82.2) [81.9, 82.6] | 10.63 | | 44482 | 35259 (79.3) [78.9, 79.6] | 3.53 | | 40920 | 34643 (84.7) [84.3, 85.0] | 6.81 |
| | 65–69 | 41056 | 28209 (68.7) [68.3, 69.2] | | 42792 | 30423 (71.1) [70.7, 71.5] | 3.49 | | 43867 | 35300 (80.5) [80.1, 80.8] | 13.08 | | 52170 | 40310 (77.3) [76.9, 77.6] | -3.86 | | 41157 | 34546 (83.9) [83.6, 84.3] | 8.54 |
| | Total | 190583 | 139409 (73.1) [73.0, 73.4] | | 185966 | 134360 (72.2) [72.1, 72.5] | -1.23 | | 181475 | 144913 (79.9) [79.7, 80.0] | 10.66 | | 204315 | 158692 (77.7) [77.5, 77.9] | -2.75 | | 186671 | 155040 (83.1) [82.9, 83.2] | 6.95 |

**Table 2. Mammography screening attendance in the Capital Region of Denmark from 2008–2017.**

| # Invited to All Rounds | # Attended All Rounds (%) | # Attended Some Rounds (%) | # Attended No Rounds (%) |
|---|---|---|---|
| 91,988 | 59,625 (64.8%) | 30,177 (32.8%) | 2,186 (2.4%) |

before. There can be several explanations for this. Women who chose to opt out of the program were not invited into subsequent rounds, causing the invited population to be selected towards participation. Women in the youngest age group had over time an increased participation, which possibly influenced their adherence as they aged and moved into older age groups. Only the 65-69-year-olds had a participation rate lower than 70% in invitation round 1. However, of all women eligible to all 5 invitation rounds, only 65% participated in all rounds, which is significantly lower than the participation rate in any of the individual rounds. This finding could be problematic, as Andersen's 2015 study found that continued regular adherence among individuals is needed for optimal protection [20].

A study of Spain's mammography screening program found that the adherence rate has decreased from 2011 to 2017 among women aged 40–69 [17]. The adherence rate in 2017 was below the recommended 70%. These results contrast with our study that found increasing participation rates in the Capital Region of Denmark well in accordance with the recommended 70%. A cross-sectional study of the Canadian mammography screening program found that participation was highest among the 60–69 age group, followed by the 50–59 age group [14].

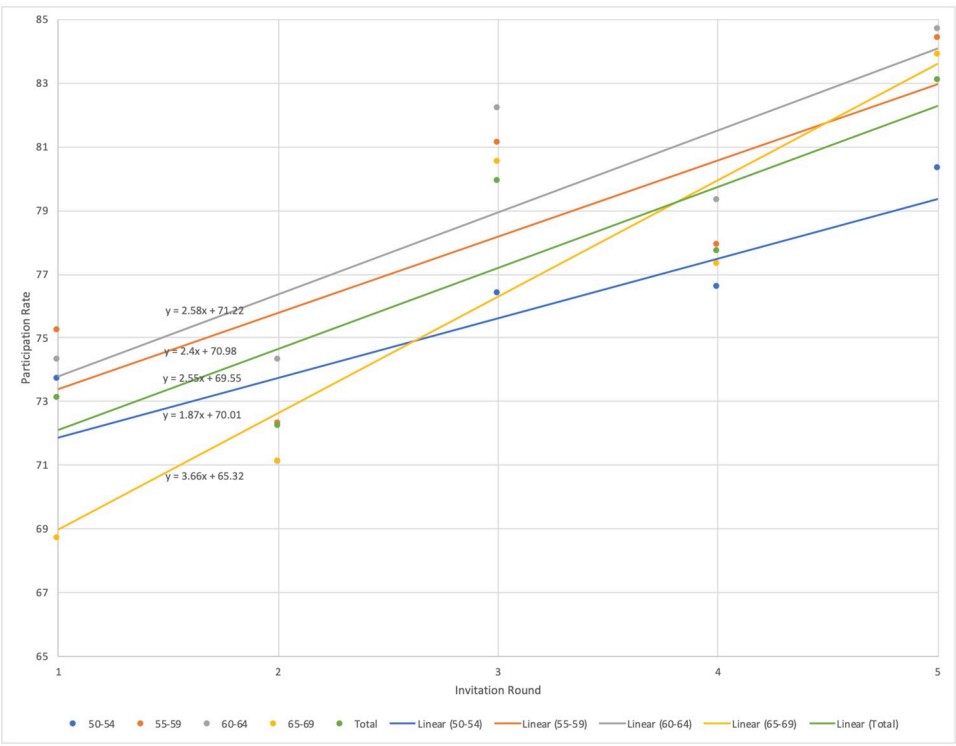

**Fig 1. Linear model of mammography screening participation rates among the four age groups and the total group in the Capital Region of Denmark from 2008–2017 (color needed).**

Overall participation rate in this study was slightly below the recommended 70%. This is in relative accordance with our results as they found that the younger women participated at a slightly lower rate than older women, but our study had a higher overall participation rate. In Lithuania, the participation rate in their national mammography screening program increased more than three-fold from 2006 to 2014 [16]. Notably, Lithuania's participation rate at the onset of the program was 20%, significantly lower than in Denmark which began with a rate over the 70% minimum. Another large study examines the participation rates for 17 European Union (EU) countries from 2004–2014 [21]. Most of the countries' participation rates have slightly decreased since their programs' inceptions, with the exception of six countries, including Denmark. The other countries with increased participation rates were Belgium, France, Czech Republic, Estonia, and Slovakia. Among these exceptions, Denmark is the only country that has consistently had a participation rate over 70% and has increased significantly since its inception. Examining an age breakdown of these other countries to determine if the age-based participation rates matched our results for the Capital Region of Denmark's program would provide further insight.

Most population-based mammography screening programs, including the Danish program, treat screening on a "one-size-fits-all" basis, where the only determinant is age. It is quite possible that age may not be the best determinant of participation. Personalized mammography screening, utilizing risk-based screening recommendations, could provide a better alternative to one-size-fits-all mammography screening programs.

A limitation of our study is that participation may be overestimated due to people opting out, creating a selection toward those who want to participate. Another limitation is the short follow-up of only 5 invitation rounds which gives room for natural variation. A strength of our study was that the capital Region Mammography Screening Register with invitation data is known to be nearly 100% complete with a very high degree of accuracy.

## Conclusions

The participation rate among all age groups has generally increased from the first round of the Capital Region of Denmark's mammography screening program to the fifth invitation round. The oldest age group experienced the largest increase in their average participation rate. Complete adherence was lower than the recommended 70%, but while high participation is a requirement for optimal benefit from screening, with age no longer being a stratifying factor, more personalized mammography screening may provide a better alternative to standardized one-size-fits-all programs.

## Acknowledgments

We wish to acknowledge the role of DIS Study Abroad, Copenhagen for research assistance.

## Author Contributions

**Conceptualization:** My von Euler-Chelpin.

**Data curation:** My von Euler-Chelpin.

**Formal analysis:** Lindsay Pett.

**Methodology:** My von Euler-Chelpin.

**Project administration:** My von Euler-Chelpin.

**Supervision:** My von Euler-Chelpin.

**Visualization:** Lindsay Pett.

**Writing – original draft:** Lindsay Pett, Becky Hollenberg, Jessica Mahoney, Jake Paz, Nathan Siu, Amanda Sun, Rachel Zhang.

**Writing – review & editing:** Lindsay Pett, My von Euler-Chelpin.

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
