## [Decision Letter · Decision Letter 0]

18 Jul 2022

PONE-D-21-29044Participation and adherence to mammography screening in the Capital Region of Denmark: The importance of age over timePLOS ONE

Dear Dr. Pett,

Thank you for submitting your manuscript to PLOS ONE; I sincerely apologise for the unusually delayed review timeframe. Your manuscript has been assessed by two reviewers, whose comments are appended below. After careful consideration, we feel that it has merit but does not fully meet PLOS ONE’s publication criteria as it currently stands. Although the reviewers comment positively on the potential importance of this work, they raise a number of concerns regarding aspects of the methodology and discussion of the results that should be addressed. Therefore, we invite you to submit a revised version of the manuscript that addresses the points raised during the review process. Please note, however, that the statements that you have provided regarding 1) the need for ethics approval and 2) review by a data protection committee satisfies our policies for research involving human participants (https://journals.plos.org/plosone/s/human-subjects-research). You therefore do not need to address the first point raised by reviewer 2, although please do include a response to this point in your rebuttal letter when you resubmit your manuscript.

We look forward to receiving your revised manuscript.

Kind regards,

Emily Chenette

Editor in Chief

PLOS ONE

Journal Requirements:

Reviewers' comments:

Reviewer's Responses to Questions

**Comments to the Author**

1. Is the manuscript technically sound, and do the data support the conclusions?

Reviewer #1: Yes

Reviewer #2: Partly

2. Has the statistical analysis been performed appropriately and rigorously? 

Reviewer #1: Yes

Reviewer #2: Yes

3. Have the authors made all data underlying the findings in their manuscript fully available?

Reviewer #1: No

Reviewer #2: No

4. Is the manuscript presented in an intelligible fashion and written in standard English?

Reviewer #1: Yes

Reviewer #2: Yes

5. Review Comments to the Author

Reviewer #1: The article deals with a very important issue. The manuscript seems technically sound to me, and the statistical analysis was performed appropriately and rigorously. The manuscript is well-written, with the exception of a couple of places where the authors used present tense when past tense was called for. The results are presented clearly in an orderly fashion and include a very informative and clearly presented table. The authors make some good points in the discussion and appropriately compare their results to those of many other studies. However, I am not clear regarding one of their attempts to explain why the most significant change occurred within the older age groups who participated to a larger extent than they did before, that being since the women aged by approximately 10 years during the study period, the higher participation is coherent with their slightly higher participation rate as younger women. I would have liked to see this idea fleshed out a little more. I agree with the authors' suggestion that personalized mammography screening utilizing risk-based screening recommendations could provide a better alternative to one-size-fits-all mammography screening programs. The authors did a good job in delineating possible limitations and a possible strength of the study. Finally, I don't see from the results the authors' statement in the conclusion that adherence was lower than the recommended 70%.

Reviewer #2: Thank you for asking me to review this manuscript.

Breast cancer is one of the most common cancer among women and important cause of death. Detection in early stage is curable and this is one of the cancers that can be screen detected. However, screening also leads to increase number of minor biopsy cases in BIRADS IV category. Voluntary participation is an important factor and adherence to screening program helps in reducing cancer mortality, this makes it an important manuscript.

I have a few observations to make

1. Authors state that this is analysis of registry data, according to me even this requires approval of the ethics committee, as authors state that Danish law does not require the approval, in that case it’s the ethical committee which should issue the waiver.

2. This is a longitudinal data over 20 year period, which is divided in 5 year age groups, during the 20 year period the participant would have moved from first group to nearly the last group, how is the movement of individual participant from one age group to other over subsequent years handled in the study?

3. Number of women participated in the program be separated from number of mammograms, as the mammograms will be repeated over period of time while number of eligible women will increase by addition of newer women becoming eligible.

4. Each women should be treated as a single case even if she underwent screening 10 times, this has not happened in this paper as all women in each round are counted this would mean a person may have been counted 10 times in 20 year period, the data should be separated.

5. There is no description of eligible population even number of women eligible at a point and overall (eligible in first year +added in second/third etc.)

6. It is not clear that the participation rates are for all eligible women or all women invited, authors should provide denominator and numerator beside percentage that is provided in text, it is important to mention how many women were eligible at a particular time

7. Has the reason for not participating recorded? Was invitation sent only once or was repeated?

8. Were any breast cancer awareness program or educational programs conducted between the study period? Were there any efforts to disseminate the importance of breast cancer screening and mammography? if so, could the increase be because of increase awareness

9. Did this increased participation resulted in increased detection (change in incidence)? Or stage shift?

10. Did this increased participation also resulted in increased rate of biopsies?

11. Most importantly did the mortality from breast cancer reduced as a result of increased participation in mammography screening?

12. What was the benefit of the program and increased participation rate that this article shows?

13. Though this study make us understand the age as important factor for participation in program, it does not inform what is the importance of this observation and also the remedies that may improve participation of younger women in breast cancer screening program (or no importance)

14. It is suggested that authors look at additional data and come out with newer findings (or benefits as sought above) that can help in improving the program and reducing the mortality from breast cancer, in its present form, the information provided in the manuscript is already known.

6. PLOS authors have the option to publish the peer review history of their article (what does this mean?). If published, this will include your full peer review and any attached files.

Reviewer #1: No

Reviewer #2: No

---

## [Author Response · Author response to Decision Letter 0]

2 Oct 2022

We thank the reviewers for their comments to our manuscript and you will find our answers below.

Have the authors made all data underlying the findings in their manuscript fully available?

Answer: We have added the following statements to the Method section:

Data from this study are stored in Statistics Denmark, which only can be accessed given the relevant data permits in accordance with GDPR and Danish Law. The data is based on personal identification numbers and open access would seriously compromise the privacy of the women included in the study.

Reviewer #1: 

However, I am not clear regarding one of their attempts to explain why the most significant change occurred within the older age groups who participated to a larger extent than they did before, that being since the women aged by approximately 10 years during the study period, the higher participation is coherent with their slightly higher participation rate as younger women.

Answer: Thank you for that comment and we agree that this is very unclear and we have rewritten the text as follows:

“Women in the youngest age group had over time an increased participation, which possibly influenced their adherence as they aged and moved into older age groups.”

I don't see from the results the authors' statement in the conclusion that adherence was lower than the recommended 70%.

Answer: This is indicated in Table 2 in the Results. However to make it clearer we have changed the sentence in Conclusion as follows:

“Complete adherence was lower than the recommended 70%, …”

Reviewer #2: 

1. Authors state that this is analysis of registry data, according to me even this requires approval of the ethics committee, as authors state that Danish law does not require the approval, in that case it’s the ethical committee which should issue the waiver.

Answer: We have been informed that the statement we have provided in the manuscript regarding the need for ethics approval and/or review by a data protection committee satisfies the journal’s policies for research involving human participants.

2. This is a longitudinal data over 20 year period, which is divided in 5 year age groups, during the 20 year period the participant would have moved from first group to nearly the last group, how is the movement of individual participant from one age group to other over subsequent years handled in the study?

Answer: Thank you for this comment. This is longitudinal data over a 10 year period and the women will thereby have moved 1 to 2 age groups over the study period. They will have been screened a maximum of 5 times. Our aim was to assess adherence over time by age group which necessitates that women are included each time they are screened and not as a single case. We hope that the reviewer can accept this explanation.

3. Number of women participated in the program be separated from number of mammograms, as the mammograms will be repeated over period of time while number of eligible women will increase by addition of newer women becoming eligible.

Answer: Please see the explanation under point 2.

4. Each women should be treated as a single case even if she underwent screening 10 times, this has not happened in this paper as all women in each round are counted this would mean a person may have been counted 10 times in 20 year period, the data should be separated.

Answer: Please see the explanation under point 2.

5. There is no description of eligible population even number of women eligible at a point and overall (eligible in first year +added in second/third etc.)

Answer: In Table 1 the absolute number of invited women and participating women are given for each age group and each invitation round. In view of our aim to assess adherence, we hope this answer is acceptable to the reviewer.

6. It is not clear that the participation rates are for all eligible women or all women invited, authors should provide denominator and numerator beside percentage that is provided in text, it is important to mention how many women were eligible at a particular time

Answer: Please see under point 5.

7. Has the reason for not participating recorded? Was invitation sent only once or was repeated?

Answer: We have not recorded the reason for not attending in this study. We have added information regarding reminders as follows:

“Women who do not show up are sent a reminder.”

8. Were any breast cancer awareness program or educational programs conducted between the study period? Were there any efforts to disseminate the importance of breast cancer screening and mammography? if so, could the increase be because of increase awareness.

In Denmark all women aged 50-69 are individually invited biennially to screening with mammography. At that time they receive information about advantages and risks with screening in order for them to make an informed choice of whether to participate or not. No regional or national campaigns have existed in the study period.

9. Did this increased participation resulted in increased detection (change in incidence)? Or stage shift?

Answer: To assess the detection rate is beyond the scope of this study, as our aim was to assess adherence to the program over time by age group.

10. Did this increased participation also resulted in increased rate of biopsies?

Answer: Please see the answer above.

11. Most importantly did the mortality from breast cancer reduced as a result of increased participation in mammography screening? 

Answer: Certainly the aim with screening is to lower the mortality from breast cancer in the population, but mortality rate is a long time measure, so in the meantime we use short time indicators, out of which one is to study participation and adherence, see European guidelines for quality assurance in breast cancer screening and diagnosis.

12. What was the benefit of the program and increased participation rate that this article shows?

Answer: The study points to the change of age as being a stratifying factor for participation to a more equal participation in all age groups, and thereby suggests that other factors might be more determining, and more personalized screening might be beneficial.

13. Though this study make us understand the age as important factor for participation in program, it does not inform what is the importance of this observation and also the remedies that may improve participation of younger women in breast cancer screening program (or no importance)

Answer: In Denmark there is no recommendation that women younger than 50 should be offered screening. Within the youngest age group are the first time invitees and therefore there are a certain group that will opt out of the program and will not be invited further. This is explained in the first paragraph of the Discussion.

14. It is suggested that authors look at additional data and come out with newer findings (or benefits as sought above) that can help in improving the program and reducing the mortality from breast cancer, in its present form, the information provided in the manuscript is already known.

Answer: We thank the reviewer for this comment and more research is certainly ongoing

---

## [Decision Letter · Decision Letter 1]

10 Jan 2023

Participation and adherence to mammography screening in the Capital Region of Denmark: The importance of age over time

PONE-D-21-29044R1

Dear Dr. Pett,

We’re pleased to inform you that your manuscript has been judged scientifically suitable for publication and will be formally accepted for publication once it meets all outstanding technical requirements.

Kind regards,

Sandar Tin Tin

Academic Editor

PLOS ONE

Reviewers' comments:

Reviewer's Responses to Questions

**Comments to the Author**

1. If the authors have adequately addressed your comments raised in a previous round of review and you feel that this manuscript is now acceptable for publication, you may indicate that here to bypass the “Comments to the Author” section, enter your conflict of interest statement in the “Confidential to Editor” section, and submit your "Accept" recommendation.

Reviewer #2: (No Response)

Reviewer #3: All comments have been addressed

2. Is the manuscript technically sound, and do the data support the conclusions?

Reviewer #2: Yes

Reviewer #3: Yes

3. Has the statistical analysis been performed appropriately and rigorously? 

Reviewer #2: I Don't Know

Reviewer #3: Yes

4. Have the authors made all data underlying the findings in their manuscript fully available?

Reviewer #2: No

Reviewer #3: (No Response)

5. Is the manuscript presented in an intelligible fashion and written in standard English?

Reviewer #2: Yes

Reviewer #3: (No Response)

6. Review Comments to the Author

Reviewer #2: I thank authors for their response however, the response to some of the questions is not satisfactory like that of Q2,3, and 4

I do not see any modifications made in the manuscript as per comments

Reviewer #3: All comments from reviewers have been addressed appropriately. It is now acceptable for publication.

7. PLOS authors have the option to publish the peer review history of their article (what does this mean?). If published, this will include your full peer review and any attached files.

Reviewer #2: No

Reviewer #3: **Yes: **Rasmi G. Nair

---

## [Editor Report · Acceptance letter]

13 Jan 2023

PONE-D-21-29044R1 

Participation and adherence to mammography screening in the Capital Region of Denmark: The importance of age over time 

Dear Dr. Pett:

I'm pleased to inform you that your manuscript has been deemed suitable for publication in PLOS ONE. Congratulations! Your manuscript is now with our production department. 

Kind regards, 

on behalf of

Dr. Sandar Tin Tin 

Academic Editor

PLOS ONE